# In-Group Love, Out-Group Hate: A Framework to Measure Affective Polarization via Contentious Online Discussions

## Abstract

Affective polarization, the emotional divide between ideological groups marked by in-group love and out-group hate, has intensified in the United States, driving contentious issues like masking and lockdowns during the COVID-19 pandemic. Despite its societal impact, existing models of opinion change fail to account for emotional dynamics nor offer methods to quantify affective polarization robustly and in real-time. In this paper, we introduce a discrete choice model that captures decision-making within affectively polarized social networks and propose a statistical inference method estimate key parameters—in-group love and out-group hate—from social media data. Through empirical validation from online discussions about the COVID-19 pandemic, we demonstrate that our approach accurately captures real-world polarization dynamics and explains the rapid emergence of a partisan gap in attitudes towards masking and lockdowns. This framework allows for tracking affective polarization across contentious issues has broad implications for fostering constructive online dialogues in digital spaces.

## CCS Concepts

• **Information systems** → **Social networks**; • **Applied computing** → *Sociology*; • **Computing methodologies** → Modeling methodologies.

## Keywords

Polarization, Twitter, COVID-19, Social Networks, Exposure

**ACM Reference Format:**
Anonymous Author(s). 2018. In-Group Love, Out-Group Hate: A Framework to Measure Affective Polarization via Contentious Online Discussions. In *Proceedings of Make sure to enter the correct conference title from your rights confirmation emai (Conference acronym 'XX)*. ACM, New York, NY, USA, 15 pages. https://doi.org/XXXXXXX.XXXXXXX

## 1 Introduction

Americans have grown increasingly divided along ideological lines. These divides extend beyond mere disagreements on policy issues to open antagonism and hostility between the two parties. This emotional divide, known as *affective polarization* [19, 20], is characterized by two key aspects: (1) the tendency for people to like and trust others from their own political party (*in-group love*) and (2) the tendency to dislike and distrust people from opposing parties (*out-group hate*) [21]. The out-group hate is a corrosive force that often undermines consensus, driving individuals to adopt a position not necessarily on its merits or because it aligns with their own beliefs, but simply to oppose the other side [9, 27]. Thus, when an issue becomes politicized, i.e., associated with a partisan identity, out-group animosity can split society, transforming seemingly mundane choices such as mask wearing and vaccination into contentious partisan divides [20]. For these reasons, affective polarization can disrupt governance in unexpected ways [17, 19].

Despite its significance, few robust methods exist to estimate the core components of affective polarization, namely, in-group love and out-group hate, in real-time. Current methods rely heavily on offline surveys [9], which are vulnerable to framing effects and respondent bias [8], and are slow. Data-driven methods usually rely on homophily [3, 31]—the tendency of similar users to form new connections—to explain emergence of polarization in social networks [12, 13]. However, these methods do not generalize well due to challenges of collecting network data, and they fail to explain divergence of partisan opinions in fully connected networks.

To address these challenges, we propose a principled framework for estimating in-group love and out-group hate using social media data. Our key contributions are as follows:

***Discrete Choice Model***: We introduce an intuitive discrete choice model of individual decision-making within an affectively polarized social network. This model includes parameters that capture the two key aspects of affective polarization: in-group love ($\alpha$) and out-group hate ($\beta$). The temporal dynamics of the model demonstrate how emotionally polarized decision-making can either unify or divide a population along party lines, depending on the values of $\alpha$ and $\beta$. We also generalize the model to account for multi-party contexts (e.g., far left, left, moderate, right, far-right), which can explain issue polarization in societies where a spectrum of ideologies exists.

***Statistical Estimation Method***: We develop a method to estimate in-group love ($\alpha$) and out-group hate ($\beta$) based on individuals' stances and those of their neighbors (both in-group and out-group) on social networks. For example, this method can analyze opinions expressed on Twitter about the effectiveness of masks in curbing the spread of COVID-19. The estimation method is derived from the discrete choice model and uses logistic regression. We also discuss variations of the method for settings where only ego nodes' stances (but not their neighbors') are available.

***Empirical Validation***: We test the proposed methods using social media posts from Twitter related to masking and lockdowns during the COVID-19 pandemic. After calibrating the model by estimating $\alpha$ and $\beta$ from data, we demonstrate a strong alignment between model predictions and real-world polarization dynamics.

Our empirical analysis of masking and lockdowns attitudes expressed online during the COVID-19 pandemic shows that a partisan gap emerged quickly on these issues, likely triggered by key

events such as the CDC masking recommendation (or co-occurring events). For the estimated parameters, the model accurately reproduces the divergence of attitudes along ideological lines, and also demonstrates that the more active partisans show higher levels of out-group hate.

By rapidly identifying emotionally polarized issues on social media through empirical estimation of in-group love and out-group hate, we can create more effective strategies that foster more constructive dialogue. News organizations and journalists can leverage this information to report on contentious issues more responsibly, thereby reducing the risk of inflaming tensions. More importantly, by recognizing emotionally divisive issues, social media platforms can implement targeted strategies in real-time to keep societal divisions from growing or being manipulated by malicious actors. This helps promote healthier participation in civic life online. The proposed model and estimation method also provides a framework for network and computational social science researchers to systematically understand manifestations of affective polarization in various contexts such as climate change, women's rights, etc.

## 2 Related Work

**Survey-based methods** The existing methods rely largely on self-reported surveys to measure affective polarization (i.e., in-group love and out-group hate) [19]. For example, the widely used *feelings thermometer* method implemented by the American National Election Study (ANES) asks democrats and republicans how warm or cold are their feelings towards their own party and the opposing party on a scale of 0 (coldest) to 100 (warmest). Other similar approaches ask about traits that people tend associate with the two parties (such as intelligence, patriotism, meanness, hypocrisy) and the comfortability to be associated with a member of the opposite party (as a friend, a neighbor, a relative, etc.) [8, 24]. Despite their wide usage, survey-based methods such as the ANES feelings thermometer have multiple limitations. One such limitation is the respondents' subjective self-interpretation of the survey questions. For example, it has been shown that people tend to think of political elites when asked about their feelings towards a political party [10]. Further, such survey-based methods have been shown to have relatively less participation from individuals who are not passionate about politics (selection bias). Additionally, the survey results are also affected by the survey mode (e.g., in-person or online) [32].

**Online Polarization** Research on social media's role in polarization has evolved, with scholars increasingly focusing on how emotional divides exacerbate it. Studies show that cross-ideological interactions on social media platforms tend to be more negative and toxic compared to exchanges between same-ideology users [23], consistent with affective polarization. Several network-based methods have been proposed that analyze network structure to detect polarization and controversy. The methods are inspired by the idea of partisan sorting, wherein social media users position themselves near same-ideology others, aligning their beliefs. Garimella et al. [13] use random-walk-based measures to analyze the structure of conversation graphs and identify topics conversations. Bonchi et al. [3] examine polarization in signed networks where positive and negative edges indicate friendly or antagonistic interactions. They use community detection to split the network by antagonistic

and friendly interactions to identify issues where disagreement exists. Similar to them, Fraxanet et al. [12] measure polarization by analyzing signed networks of online interactions. They quantify polarization as the interplay between 1) *antagonism*, which reflects the level of negative interactions or hostility within a community, and 2) *alignment*, measured by how much interactions are structured along a primary division, or fault line, within the community. They identify issues that drive polarization (high antagonism and alignment) and conflicts that do not reinforce societal divisions.

Polarization grew during the COVID-19 pandemic, with partisan divides emerging across multiple issues like virus origins, mitigation strategies (e.g., social distancing, lockdowns, masking), and vaccine mandates [22, 29]. Research demonstrated that political ideology significantly influenced adherence to health guidelines and mandates [15, 17]. Political elites contributed to some of the polarization of public opinion and behavior during the pandemic's early stages [16] by focusing on separate issues.

***How the proposed method differs from existing methods.*** This paper introduces an interpretable logistic model applied to social media data to estimate affective polarization. The model expresses the log-odds of a person adopting a stance (e.g., mask-wearing) as a linear function of two time-varying covariates: the prevalence of the stance among in-group and out-group connections at the previous time point. Social media data on individual stances reveal these two covariates. The proposed model can then be used to estimate in-group love and out-group hate, which quantify the relative importance of the two covariates. Unlike survey-based methods, this data-driven approach accounts for social network structure and temporal dynamics, offering statistical guarantees and applicability to various issues like vaccines and climate change. It is objective, scalable, interpretable, and compatible with modern tools like LLMs.

The binary choice model we rely on for estimating affective polarization is a generalized version of the model presented in the theoretical study in [27]. This generalized model we propose can better replicate the real-world observations and is also amenable to logistic regression for estimating the in-group love $\alpha$ and the out-group hate $\beta$ parameters (with statistical guarantees). The model presented in [27] follows as a special case of our model.

Our work makes additional contributions to the state of the art. Researchers have not explained how opinions on novel issues raised by the COVID-19 pandemic became polarized so quickly, nor have they examined the role of affective polarization in the emergence of the partisan gap in COVID-19 attitudes.

## 3 Modeling Affective Polarization

This section introduces a dynamical model of individual decisions in an affectively polarized society, separating the effects of in-group love and out-group hate.

### 3.1 The Binary Choice Model

**Context:** We consider an undirected social network $G = (V, E)$ where each node (individual) $v \in V$ has two binary attributes: a static attribute $R(v) \in \{0, 1\}$ representing its identity or aspect of an identity such as ideology or political affiliation, and a dynamic

attribute $H_k(v) \in \{0, 1\}$ that evolves over time $k = 0, 1, 2, \ldots$. The node $v$ is labeled as red if $R(v) = 1$ (and blue otherwise). The dynamic attribute $H_k(v)$ reflects the individual $v$'s stance (or choice) from among the two available alternatives at time $k$, such as wearing a mask or not. The initial stances (i.e., $H_k(v)$ for each $v \in V$ at time $k = 0$) may be assigned deterministically or randomly.

Each individual can observe the stances of their network neighbors, regardless of their own or the neighbor's affiliation. We refer to individual's same-affiliation neighbors as their *in-group* and opposite affiliation neighbors as their *out-group*.

**Quantifying the In-Group and Out-Group Influences:** In order to define the evolution of stances of individuals, we first need to formally specify how each individual is influenced by the stances of their in-group and out-group neighbors. For this purpose, let $\Delta_k^{in,1}(v), \Delta_k^{out,1}(v)$ denote measures that quantify the prevalence of stance-1 (with respect to the stance-0) among the in-group neighbors and the out-group neighbors of $v \in V$, respectively. Similarly, $\Delta_k^{in,0}(v), \Delta_k^{out,0}(v)$ denote the prevalence of stance-0 (with respect to the stance-1) among the in-group and out-group neighbors of $v$, respectively. The application context and the available data granularity could dictate the exact definitions of those measures of in-group and out-group influences. Below are two examples.

*Definition 1. Net number of individuals with a stance normalized by degree:* $\Delta_k^{in,1}(v)$ is defined as the difference between the number of in-group neighbors of $v$ with stance-1 and stance-0, normalized by the degree of $v$ (number of neighbors of the $v$), and $\Delta_k^{in,0}(v) = -\Delta_k^{in,1}(v)$. The quantities for the out-group are defined similarly. For example, consider a blue node $v$ with 70 out of 100 blue neighbors (in-group connections) wearing masks (stance 1) and 7 out of 10 red-neighbors (out-group connections) wearing masks. Under this first definition of influence, we get $\Delta_k^{in,1}(v) = (70 - 30)/110 = 40/110, \Delta_k^{in,0}(v) = -40/110$ and $\Delta_k^{out,1}(v) = (7 - 3)/110 = 4/110, \Delta_k^{out,0}(v) = -4/110$. This approach takes into account the number of individuals with stance-1 and not just the fraction in each group (i.e., a larger number of individuals is likely to exert more influence).

*Definition 2. Net fraction of neighbors in each group with a stance:* Alternatively, $\Delta_k^{in,1}(v)$ is defined as the difference between fraction of in-group neighbors of $v$ with stance-1 and fraction of in-group neighbors of $v$ with stance-0, and $\Delta_k^{in,0}(v) = -\Delta_k^{in,1}(v)$. For the same example as before, this second approach would yield same magnitude of the in-group and out-group effects from masking i.e., $\Delta_k^{in,1}(v) = \Delta_k^{out,1}(v) = 0.7 - 0.3 = 0.4$ and $\Delta_k^{in,0}(X_k) = \Delta_k^{out,0}(v) = -0.4$. Thus, unlike the first approach, given the fraction of individuals in each group of neighbors who are masking and non-masking, the actual numbers do not matter.

The formal versions of the above definitions are given in the Appendix. For the remainder of this paper, we use the first definition. Results related to Definition 2 are provided in the Appendix.

**Dynamics of the Stances:** At each time step $k$ (where $k = 0, 1, 2, \ldots$), a node $X_k \in V$ is sampled uniformly to observe the stances of its neighbors and then update its own stance as follows. Let,

$$p_{X_k,k}(1|0) = \mathbb{P}(H_{k+1}(X_k) = 1 | H_k(X_k) = 0) \quad (1)$$
$$p_{X_k,k}(0|1) = \mathbb{P}(H_{k+1}(X_k) = 0 | H_k(X_k) = 1) \quad (2)$$

denote the probabilities with which the node $X_k$ switches its choice at time $k + 1$. Those transition probabilities are modeled using the following logistic functions,

$$p_{X_k,k}(1|0) = \frac{1}{1 + \exp\left[-\left(\alpha\Delta_k^{in,1}(X_k) - \beta\Delta_k^{out,1}(X_k) - \delta\right)\right]}$$
$$p_{X_k,k}(0|1) = \frac{1}{1 + \exp\left[-\left(\alpha\Delta_k^{in,0}(X_k) - \beta\Delta_k^{out,0}(X_k) - \delta\right)\right]}, \quad (3)$$

where $\alpha, \beta, \delta \geq 0$ are fixed model parameters. In the event that the node $X_k$ doesn't switch choices, it stays with the same choice at time $k + 1$, i.e., $H_{k+1}(X_k) = H_k(X_k)$.

**Discussion of the Model:** In the above model, the parameters $\alpha, \beta$ quantify the strengths of in-group love and out-group hate, respectively. For example, consider a scenario where the randomly selected node at time $k$, $X_k$, belongs to the blue group (i.e., $R(X_k) = 0$) and is not wearing a mask (i.e., holds stance-0 or $H_k(X_k) = 0$). In order to decide whether to adopt stance-1 (wear a mask) at the next time step, $k + 1$, $X_k$ focuses on two factors: the prevalence of stance-1 among the in-group (blue) neighbors (quantified by $\Delta_k^{in,1}(X_k)$) and the prevalence of stance-1 among out-group (red) neighbors (quantified by $\Delta_k^{out,1}(X_k)$). In particular, notice from Eq. (3) that the log-ratio of adopting stance-1 to staying with stance-0 can be expressed as,

$$\alpha\Delta_k^{in,1}(X_k) - \beta\Delta_k^{out,1}(X_k) - \delta = \log\left(\frac{p_{X_k,k}(1|0)}{1 - p_{X_k,k}(1|0)}\right)$$
$$= \text{logit}\left(p_{X_k,k}(1|0)\right).$$

A graphical illustration is given in the Appendix Fig. 9. If the in-group neighbors are more inclined towards stance-1 (i.e., larger $\Delta_k^{in,1}(X_k)$), it encourages $X_k$ to adopt stance-1 at the next time step (i.e., the probability of switching to stance-1, $p_{X_k,k}(1|0)$, is larger). On the other hand, if the in-group neighbors are less inclined to wear masks (i.e., smaller $\Delta_k^{in}(X_k)$), it discourages $X_k$ from masking. The out-group has the opposite effect: a higher tendency among the out-group to wear masks (i.e., larger $\Delta_k^{out,1}(X_k)$) provokes $X_k$ to resist masking (reflecting out-group animosity) and vice-versa. Thus, the decision of $X_k$ is influenced by the interplay of these two social tendencies. The relative strengths of the in-group love and out-group hate are quantified by the parameters $\alpha$ and $\beta$, respectively. For example, $\beta > \alpha$ indicates that people pay more attention to the out-group than the in-group when making decisions. The inertia $\delta$ models the tendency to resist changes in behavior. In particular, larger values of $\delta$ implies that a large collective influence of the in-group and out-group is needed for $X_k$ to switch its stance.

The proposed model decouples the effects of in-group love and out-group hate (e.g., high, equal, low out-group hate compared to in-group love), and help understand how their interplay shape the dynamics of people's stances. Additionally, it takes into account the fact that people may be exposed to more in-group individuals (due to homophily of the political ideology in the network $G = (V, E)$)

as well as the asymmetry of the presence of the two stances within the in-group and out-groups of individuals.

## 3.2 Dynamics of the Discrete Choice Model

The evolution of the model presented in Sec. 3.1 is stochastic since the node that updates its stance at each time instant is chosen at random. However, the stochastic dynamics can be meaningfully approximated in a mean-field manner when the number of nodes are large and the network is fully connected. In particular, let $\theta^{\mathcal{B}}(t), \theta^{\mathcal{R}}(t)$ denote the fraction of blue nodes with stance-1 and fraction of red-nodes with stance-1 at (continuous) time $t$, respectively. Further, let $r \in (0, 1)$ be the fraction of red nodes in the network. The influence measures defined earlier ($\Delta_k^{in,1}(X_k), \Delta_k^{out,1}(v), \Delta_k^{in,0}(v), \Delta_k^{out,0}(v)$) can then be written in terms of $\theta^{\mathcal{B}}(t), \theta^{\mathcal{R}}(t)$ and $r$. For example, for any blue node, $\Delta_k^{in,1}(v) = (1 - r)\left(2\theta^{\mathcal{B}}(t) - 1\right)$ under the first influence measure (net number of individuals with a stance normalized by degree) and $\Delta_k^{in,1}(v) = \left(2\theta^{\mathcal{B}}(t) - 1\right)$ under the second influence measure (fraction of neighbors in each group with a stance). Consequently, the transition probabilities in Eq. (3) can be expressed in terms of $\theta^{\mathcal{B}}(t), \theta^{\mathcal{R}}(t)$ and $r$. In particular, let $p_\theta^{\mathcal{B}}(1|0), p_\theta^{\mathcal{B}}(0|1)$ (resp. $p_\theta^{\mathcal{R}}(1|0), p_\theta^{\mathcal{R}}(0|1)$) denote the transition probabilities given in Eq. (3) when $X_t$ is a blue (resp. red) node in a fully connected graph. Under the first influence measure defined earlier (net number of individuals with a stance normalized by degree), they can be expressed in terms of $\theta^{\mathcal{B}}(t), \theta^{\mathcal{R}}(t)$ and $r$ as,

$$\text{logit}\left(p_\theta^{\mathcal{B}}(1|0)\right) = \alpha(1-r)\left(2\theta^{\mathcal{B}}(t) - 1\right) - \beta r\left(2\theta^{\mathcal{R}}(t) - 1\right) - \delta,$$

$$\text{logit}\left(p_\theta^{\mathcal{B}}(0|1)\right) = \alpha(1-r)\left(1 - 2\theta^{\mathcal{B}}(t)\right) - \beta r\left(1 - 2\theta^{\mathcal{R}}(t)\right) - \delta,$$

$$\text{logit}\left(p_\theta^{\mathcal{R}}(1|0)\right) = \alpha r\left(2\theta^{\mathcal{R}}(t) - 1\right) - \beta(1-r)\left(2\theta^{\mathcal{B}}(t) - 1\right) - \delta,$$

$$\text{logit}\left(p_\theta^{\mathcal{R}}(0|1)\right) = \alpha r\left(1 - 2\theta^{\mathcal{R}}(t)\right) - \beta(1-r)\left(1 - 2\theta^{\mathcal{B}}(t)\right) - \delta.$$

The transition probabilities under second measure (fraction of neighbors in each group with a stance) can be obtained by simply removing $r$ from the above expressions (since the group sizes do not matter). Then, the evolution of $\theta(t) = \left[\theta^{\mathcal{B}}(t), \theta^{\mathcal{R}}(t)\right]$ on a fully connected network can be represented using the following differential equation

$$\begin{bmatrix} \dot{\theta}^{\mathcal{B}}(t) \\ \dot{\theta}^{\mathcal{R}}(t) \end{bmatrix} = \begin{bmatrix} \left(1 - \theta^{\mathcal{B}}(t)\right)p_\theta^{\mathcal{B}}(1|0) - \theta^{\mathcal{B}}(t)\,p_\theta^{\mathcal{B}}(0|1) \\ \left(1 - \theta^{\mathcal{R}}(t)\right)p_\theta^{\mathcal{R}}(1|0) - \theta^{\mathcal{R}}(t)\,p_\theta^{\mathcal{R}}(0|1) \end{bmatrix}, \quad (4)$$

where $\dot{\theta}^{\mathcal{B}}(t), \dot{\theta}^{\mathcal{R}}(t)$ are the rates of change of $\theta^{\mathcal{B}}(t), \theta^{\mathcal{R}}(t)$, respectively.

The trajectories of the differential equation in Eq. (4) can easily be obtained numerically.[1] Fig. (1) shows example scenarios where both red and blue groups initially have the same prevalences of dynamic attribute (i.e., $\theta^{\mathcal{B}}(0) = \theta^{\mathcal{R}}(0)$). Several observations can be made from Fig. 1.

---

[1]Using the Euler method, we can get the discretized trajectory as,

$$\theta^{\mathcal{B}}(k\epsilon + \epsilon) = \theta^{\mathcal{B}}(k\epsilon) + \epsilon\dot{\theta}^{\mathcal{B}}(k\epsilon) \quad (5)$$

$$\theta^{\mathcal{R}}(k\epsilon + \epsilon) = \theta^{\mathcal{R}}(k\epsilon) + \epsilon\dot{\theta}^{\mathcal{R}}(k\epsilon) \quad (6)$$

at time steps $t = k\epsilon$ where $k = 0, 1, 2, \ldots$ for some small $\epsilon > 0$.

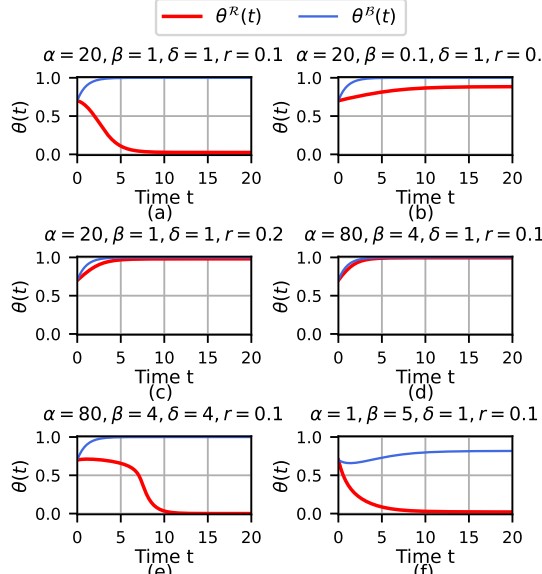

Figure 1: Example trajectories of the $\theta(t) = \left[\theta^{\mathcal{B}}(t), \theta^{\mathcal{R}}(t)\right]$ on a fully connected graph based on Eq. (4) (under the first definition of peer influence) for various parameter configurations. In each case, it is assumed that $\theta^{\mathcal{B}}(0) = \theta^{\mathcal{R}}(0)$ i.e., both groups initially have the same prevalence of the dynamic attribute.

First, comparing Fig. (1)(a) and Fig. (1)(b) shows that decreasing the out-group hate $\beta$ (compared to in-group love $\alpha$) facilitates consensus where both groups largely adopt the same stance over time (i.e., $\theta^{\mathcal{B}}(0) \approx \theta^{\mathcal{R}}(0) \approx 1$ or $\theta^{\mathcal{B}}(0) \approx \theta^{\mathcal{R}}(0) \approx 0$). Importantly, the values of $\alpha, \beta$ matter and not just their ratio. For example, Fig. (1)(a) and Fig. (1)(d) both have the same $\alpha$ to $\beta$ ratio, and yet correspond to two different outcomes (partisan polarization and consensus).[2]

Second, more balanced group sizes (i.e., $r$ closer to 0.5) facilitate consensus as seen by comparing Fig. (1)(a) and Fig. (1)(c). Comparing Fig. (1)(d) with Fig. (1)(e) shows how inertia (while all other parameters kept same) can affect the dynamics of polarization.

Fig. (2) shows several scenarios where the two groups have different initial states (i.e., $\theta^{\mathcal{B}}(0) \neq \theta^{\mathcal{R}}(0)$). In particular, Fig. (2)(b) shows an example of a cross-over where the minority (red) group ends up largely giving up the stance-1 despite that being initially more prevalent compared to the blue group. Fig. (2)(c) shows an example of the minority red group (red) reversing the trend when the two groups are about to reach consensus.

Interestingly, under the second influence measure specified by Definition 2, both groups follow an identical trajectory when the initial states are same for both groups. This can be seen from Fig. 10 in the Appendix. Intuitively, since the initial fractions of stances

---

[2]In the model proposed in [27], only $\alpha/\beta$ and $r/(1-r)$ determine the outcomes over a long period of time. The long-term outcomes of the model proposed in [27] is limited to exact consensus (where $\theta^{\mathcal{B}}(t) = \theta^{\mathcal{R}}(t) = 0$ or $\theta^{\mathcal{B}}(t) = \theta^{\mathcal{R}}(t) = 1$ as $t$ goes to infinity), exact partisan polarization (where $\theta^{\mathcal{B}}(t) = 1, \theta^{\mathcal{R}}(t) = 0$ or $\theta^{\mathcal{B}}(t) = 0, \theta^{\mathcal{R}}(t) = 1$ as $t$ goes to infinity) or non-partisan polarization where each group will be split evenly (where $\theta^{\mathcal{B}}(t) = \theta^{\mathcal{R}}(t) = 0.5$ as $t$ goes to infinity). In contrast, the generalized model is able to capture scenarios in between as seen from Fig. (1) and Fig. (2).

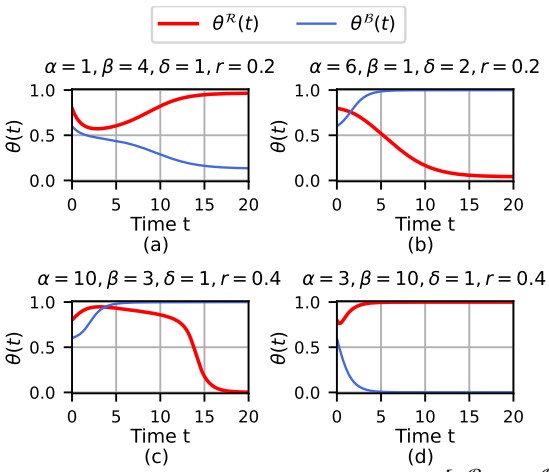

**Figure 2: Example trajectories of the $\theta(t) = \left[\theta^{\mathcal{B}}(t), \theta^{\mathcal{R}}(t)\right]$ on a fully connected graph based on Eq. (4) (under the first definition of peer influence) for various parameter configurations. Unlike Fig. 1, the two groups initially have different prevalences of the dynamic attribute.**

are the same in each group, both the in-group and out-group effects for each group remains symmetric and therefore the trajectories are identical. When the initial conditions for the two groups are different, the dynamics under Definition 2 can show a rich array of phenomena as seen from Fig. (11) in the Appendix.

As Fig. 1 and Fig. 2 illustrate, the proposed model can replicate a wide-array of phenomena observed in real-world even on a fully connected network. Further, the proposed model and its analysis can be extended to multi-party systems (beyond two parties) as shown in Appendix D.

## 4 Estimating Affective Polarization

We illustrate how the parameters of the above model can be estimated from social media data. We present two approaches for two levels of granularity of the available data: stances (dynamic attribute) and parties (static attribute) of a set of sampled nodes as well as their neighbors are known (case 1) and stances and party of a set of randomly sampled nodes available (case 2).

***Case 1: Stances of Sampled Egos and their Neighbors are Known*** When the stance as well as neighbor influence measures are known for a sample of ego nodes $v_i, i = 1, 2, \ldots, n$, model parameters can be estimated without any further assumptions. Specifically, we assume that the peer influences (based on in-group and out-group neighbors with each stance) at some time instant $k_i$, $\Delta_{k_i}^{in,1}(v_i), \Delta_{k_i}^{out,1}(v_i)$, $\Delta_{k_i}^{in,0}(v_i), \Delta_{k_i}^{out,0}(v_i)$, are known for each node $v_i, i = 1, 2, \ldots, n$. Further, the stance of each ego node in the sample $v_i, i = 1, 2, \ldots, n$ at two consecutive time instants, $k_i$ and $k_i + 1$ (i.e., $H_{k_i}(v_i), H_{k_i+1}(v_i)$), are also known. Then, logistic regression can be utilized to estimate the parameters $\alpha, \beta, \delta$ as shown in Algorithm 1 (Refer Appendix B.3). It leverages the logistic model Eq. 3, ensuring that all theoretical properties of logistic regression via maximum likelihood are preserved. A key practical point is that the same node can appear in observations at different time intervals, allowing monitoring of

a few nodes' stances and influence over time as sufficient input for Algorithm 1. Additionally, no assumptions about the network structure are required to estimate $\alpha, \beta, \delta$ with theoretical guarantees.

***Case 2: Only Stances of Sampled Egos are Known*** While collecting stances from a random set of ego nodes over time is feasible, obtaining peer influence measures is difficult due to the need for network structure. In such cases, the random set of ego nodes can be treated as a sample from a fully connected network. For Algorithm 1, this means that the influence measures $\Delta_{k_i}^{in,1}(v_i), \Delta_{k_i}^{out,1}(v_i)$, $\Delta_{k_i}^{in,0}(v_i), \Delta_{k_i}^{out,0}(v_i)$ are calculated by viewing each $v_j, j \neq i$ in the sample as a neighbor of $v_i$.

## 5 Empirical Validation

We validate the model empirically, by calibrating its parameters on real-world discussions about contentious issues on Twitter.

### 5.1 Data and Issue Detection

We use publicly available data consisting of 1.4 billion tweets related to the COVID-19 pandemic posted between January 21, 2020, and November 4, 2021 [4]. These tweets were collected using pandemic-relevant keywords and geolocated within the US with Carmen [7], a geo-location tool for Twitter data. Carmen leverages tweet metadata, including "place" and "coordinates" objects, as well as mentions of locations in users' bios, to assign tweets to specific locations at state and county level (see Appendix E.3). We restrict our focus to tweets between January 21, 2020 and January 1, 2021. This leaves us with 230M tweets from 8.7M users in the US.

Prophylactic measures like stay-at-home orders and masking recommendations dominated the early online discussions about the pandemic [5, 30], growing contentious [23, 28]. The *masking* issue encompasses posts about the use of face coverings, mask mandates, mask shortages, and anti-mask sentiments. The *lockdowns* issue includes discussions of state and federal mitigation efforts, such as quarantines, stay-at-home orders, business closures, reopening strategies, calls for social distancing and access to transportation. To identify posts relevant to these issues, we employ a weakly-supervised approach which mines Wikipedia pages related to each issue for relevant keywords and phrases [28]. After manually verifying the extracted terms, we filter the tweets for posts that mention these issues. We identify 11.4M tweets as relevant to masking and 8.3M tweets as relevant to lockdowns.

### 5.2 Stance Classification

Pro-masking tweets advocate masks as an effective public health measure, while anti-masking tweets argue masks harm physical and mental well-being and infringe on personal liberty. Neutral tweets often include news or unrelated content. Similarly, pro-lockdown tweets emphasize lockdowns as essential for controlling COVID-19 and protecting healthcare systems, whereas anti-lockdown tweets highlight economic, mental health harms, and threats to personal freedoms. Neutral tweets on lockdowns typically share news or related information without a clear stance. See Table 3 in the Appendix for examples.

We finetune a LLaMA 3.1-8B Instruct model with Low-Rank Adaptation (LoRA)[18], to identify the stance expressed in a tweet.

The training data for finetuning comes from [14], which provides a set of annotated tweets related to masking and lockdowns. Annotations categorize the stance on each issue as: favor, against, and neutral/irrelevant. Glandt et al. [14] offers 1,921 tweets annotated for masking stances and 1,717 tweets for lockdowns, from which we utilize 80% for training and allocate 10% each for validation and testing. LoRA is particularly advantageous for small datasets as it enables effective fine-tuning of large pre-trained models while minimizing the number of parameters that need to be adjusted during training, allowing the model to adapt without extensive retraining. This efficient parameterization helps maintain the foundational knowledge embedded in the pre-trained model, thereby, facilitating faster convergence. More specifically, we use tweets as input to the model with the prompt - `What is the stance expressed towards masking (resp. lockdowns mandates) in the following tweet?` and the stances for the corresponding tweet from [14] as the ground-truth stance. We then use the finetuned models to infer stances for all $11.4M$ masking-related tweets and $8.3M$ lockdown-related tweets. We find $6.7M$ masking-related tweets favor masking (pro-masking), $1.2M$ are against (anti-masking), and $3.2M$ are neutral/not-relevant; of the $8.3M$ lockdown-related tweets, $2.1M$ favor lockdowns, $1.4M$ are against and $4.8M$ are neutral/not-relevant. Stance classifier performance on the held out test sets is highly reliable with F1-scores of 0.91 and 0.85 for masking and lockdowns. Masking stances are also highly correlated at state level with off-line masking survey conducted by New York Times (Pearson $r = 0.63$, see Appendix E.1).

## 5.3 User Ideology Classification

To identify ideology, we use a two-phase approach. First, we reference a curated list of political elites (e.g., politicians, pundits) with ideological leanings of $-1$ for liberals and $1$ for conservatives. We analyze frequent retweet interactions between regular users and these elites, creating a bipartite network and excluding interactions with fewer than 10 occurrences. This yields roughly 3 million interactions between 92,000 users and 2,200 elites. Following [1], we map users to latent ideology space by positioning ideologically similar accounts closer based on retweet behavior, as retweets often reflect agreement.

Barberá [1] leverages two other indicators—political interest and elite popularity— in addition to retweet interactions to quantify user ideology in latent space. Political interest can be estimated as the total number of tweets, retweets, replies and quoted tweets that non-elites generate over time. Elite popularity is quantified as the in-degree of elite nodes in the bipartite network. This accounts for the variation in activity of non-elites and the popularity of elites (realDonaldTrump maybe more popular than SenatorRomney). After log-transforming both features, we model the probability of user $i$ retweeting elite $j$ as a logit:

$$p(y_{i,j} = 1 \mid \lambda_j, \eta_i, \gamma, \phi_j, \theta_i) = logit^{-1}(\lambda_j + \eta_i - \gamma * (\phi_j - \theta_i)^2),$$

where $\lambda_j, \eta_i$ represent the popularity of elite node $j$ and political interest of user $i$ respectively, and $\phi_j$ and $\theta_i$ denote the ideology of the elite $j$ and user $i$ in the latent ideological space. Parameter $\gamma$ is a regularizing constant and $(\phi_j - \theta_i)^2$ quantifies the distance between elite $j$ and user $i$ in the latent space. The goal is to maximize the

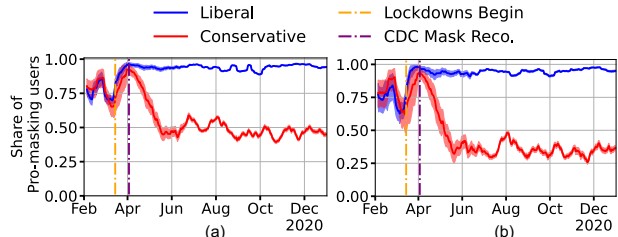

**Figure 3: Masking stances over time. Share of liberal and conservative users expressing pro-masking stance for (a) all users and (b) partisans.**

likelihood of the interaction between elite $j$ and user $i$ while minimizing the distance between them. Maximum likelihood estimation is intractable given that we have to estimate five parameters over $94K$ accounts. Instead, we use Bayesian inference. We use normal priors for $\lambda, \eta, \theta, \phi$ and half-normal prior for $\gamma$ (need for positive regularizing constant). We set initial values for $\lambda, \eta$ to be the log-transformed values for elite popularity and user political interest estimated from our dataset. We set the initial values for ideological leaning of elites $\phi$ to be the ones we obtained from [26]. We then use PyMC3 to run the No-U-Turn Sampler (NUTS) to sample from the posterior distribution of the model parameters, which include $\lambda$, $\eta, \gamma, \theta$, and $\phi$. The sampling process estimates the joint posterior distribution $(p(\lambda, \eta, \gamma, \theta, \phi \mid y))$, given the observed binary outcomes $y$. We draw a total of 1000 samples under four chains, after a warm-up period of 500 iterations for tuning. We achieve significant speedup by running the sampling on a RTXA6000 GPU node by leveraging the Numpyro Python library.

This method enables us to estimate continuous ideology scores for all users in the bipartite network of elite interactions. We call these users *political partisans*. We compare ideology scores to those estimated using the follower network-based approach in [1] for an overlap of $40K$ users and find a strong agreement (Pearson $r = 0.89$). After binarizing scores with a threshold of 0 (liberal $\leq 0$, conservative $> 0$), we achieve an F1-score of 0.95 for ideology classification.

However, most users in our dataset do not interact with political elites. In the second phase, we extend ideology classification to all users via supervised fine-tuning of the LLaMA 3.1-8B Instruct model. We compile all tweets from 8.7 million users between January 21, 2020, and December 31, 2020, creating a document for each user that aggregates all their tweets. We then use the prompt - `What is the political leaning expressed in these tweets?` We randomly select 80% of the 94K users as a training set, using their tweets as input to the model and the estimated ideology from the first phase as the output. After fine-tuning the model, we test it on the remaining 20% and achieve an F1-score of 0.98. Using the fine-tuned model, we then infer the ideology of all 8.7 million users in our dataset, resulting in approximately $6.6M$ liberal accounts and $2.1M$ conservative accounts. We perform further validation by comparing the share of conservative Twitter users at the state-level to the 2020 Federal Election Republican vote share for the state (Refer Appendix E.1).

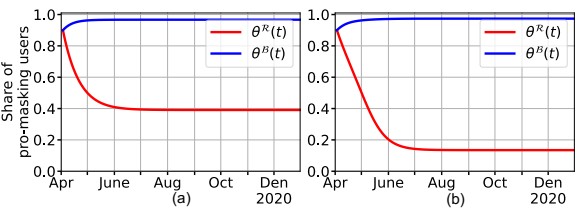

Figure 4: Lockdown stances over time. Share of liberal and conservative users expressing pro-lockdown stance for (a) all users and (b) partisans.

Table 1: Affective Polarization Parameters Estimated via Logistic Regression from Social Media Data.

| Issue | | All users | Pseudo-R2 | Partisans | Pseudo-R2 |
|---|---|---|---|---|---|
| Masking | $\alpha$ | 3.75±0.008 | | 5.11 ± 0.018 | |
| | $\beta$ | 0.25±0.005 | 0.31 | 0.63± 0.007 | 0.38 |
| | $\delta$ | 0.63±0.003 | | 0.28 ± 0.005 | |
| Lockdowns | $\alpha$ | 3.76 ± 0.020 | | 5.08± 0.032 | |
| | $\beta$ | 0.75± 0.017 | 0.15 | 1.05± 0.027 | 0.23 |
| | $\delta$ | 0.91 ± 0.004 | | 0.80 ± 0.007 | |

A user is classified as pro-masking (or pro-lockdowns) if their average stance across all tweets on the issue exceeds 0.5, otherwise as anti. Figs. 3 (a) and (b) show the share of pro-masking users over time by user ideology, for all users and partisans. Before the CDC's April 3, 2020 masking recommendation, liberals and conservatives supported masking at similar rates. Afterward, a partisan gap emerged, with a sharp decline in conservative support, especially among conservative partisans (Figures 3(b)).

Figs. 4(a) and (b) show the share of pro-lockdown users split by ideology for all users and partisans. Like masking, the novel issue of lockdowns quickly became polarized. Before U.S. stay-at-home orders (around March 15, 2020), there were no significant differences in attitudes of liberals and conservatives. Afterward, pro-lockdown sentiment sharply declined among conservatives, especially partisans. Liberals initially showed a gradual decline in support for lockdowns until June 2020, after which their pro-lockdown attitudes increased sharply.

## 5.4 Model Calibration via Parameter Estimation

We use the method described under case 2 in Section 4 to estimate affective polarization parameters. which assumes a fully connected network. Under this assumption, the in-group of liberal users includes all other liberals, and their out-group includes all conservatives, and vice versa. The number of active users varies over time, as not all users are consistently engaged on Twitter, resulting in sparse time series. To address this challenge, we divide the timeline into 24 intervals of 15 days. A user is considered active if they post within that interval. We ensure that the same users are active in two consecutive intervals; e.g., intervals $t = i$ and $t = i + 1$ share the same users, which may be different from users active during intervals $t = i + 1$ and $t = i + 2$. This allows us to assess the changes

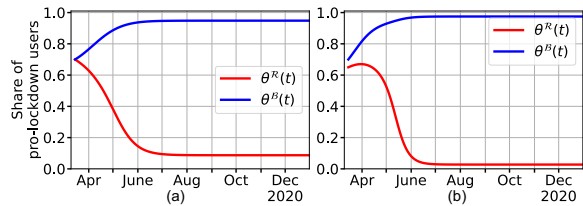

Figure 5: Plotting trajectories of issue positions on masking for (a) all users and (b) political partisans.

Figure 6: Plotting trajectories of issue positions on lockdowns for (a) all users and (b) political partisans.

in user stances between two consecutive intervals based on the stances of the user's in-group and out-group members.

The results of the estimated parameters using the above approach (see Appendix E.2 for more details) are shown in Table 1. We observe that for both issues—masking and lockdowns—$\alpha > \beta$, suggesting that opinions within a user's in-group have a greater impact on their own stance than changes in views among the out-group. However, positive $\beta$ values suggest that out-group stances influence users' own positions on masking and lockdowns. We observe that the estimated values of $\alpha$ and $\beta$ are higher for political partisans compared to all users, suggesting that their positions are more strongly influenced by in-group favoritism and out-group hostility. Additionally, the relatively high Pseudo-R2 values for both issues and user groups indicate that these parameters account for a significant portion of the variation in stance transitions over time.

## 5.5 Model Solutions with Estimated Parameters

Given three estimated parameters ($\alpha$, $\beta$, $\delta$), and three parameters measured from data— $r$, $\theta^{\mathcal{B}}(0)$, $\theta^{\mathcal{R}}(0)$, where $r$ denotes the fraction of conservative users, and $\theta^{\mathcal{B}}(0)$, $\theta^{\mathcal{R}}(0)$ are the initial share of liberals ($\mathcal{B}$) and conservatives ($\mathcal{R}$) who are pro-masking (resp. lockdowns)—we solve the ODE in Eq. 4 numerically to obtain the trajectories of the system. We set March 15, 2020 as the initial time ($t = 0$) for the lockdowns issue ($\theta^{\mathcal{B}}(0) = \theta^{\mathcal{R}}(0) = 0.7$) and April 3, 2020 for the masking issue ($\theta^{\mathcal{B}}(0) = \theta^{\mathcal{R}}(0) = 0.9$). We chose these starting points as they denote key events—CDC masking recommendation on April 3, 2020 and lockdown orders on March 15, 2020—following which the attitudes of liberals and conservatives diverged. The share of conservatives among all users at $t = 0$ discussing masking (resp. lockdowns) was 18% (resp. 43%). The share of conservatives among all partisans was 34% and 49% for masking and lockdowns respectively. Numerically solving the differential equation in Eq. (4) using Euler's method allows us to estimate, $\dot{\theta}^{\mathcal{B}}(t)$ and $\dot{\theta}^{\mathcal{R}}(t)$, which denote the changes in share of users who are liberal pro-masking (resp. lockdowns) and conservative pro-masking (resp. lockdowns) at small time steps ($t = 0.01$). We calculate the

trajectories from $t = 0$ to 100. Figs.5 and 6 show results with the time axis scaled to align with empirical data, with one time step corresponding to seven days.

The model trajectories shown in Figs. 5&6 capture real-world trajectories in Figs. 3&4. In particular, the model correctly reproduces a larger gap among active partisans than among all users, and also higher approval of masking among conservatives unlike lockdowns. However, the model overestimates approval of lockdowns among liberals, compared to real-world data Figs.3 and 4). The discrepancy may stem from our simplifying assumptions, such as a fully connected network and all users make synchronous transitions.

Prior studies [2, 3, 6, 31] suggest that homophily alone may explain the development of issue polarization. Along with estimating homophily on individual issue stance, captured by in-group love ($\alpha$), we also estimate the impact of the out-group ($\beta$). The parameter $\delta$, as previously discussed, measures the user's resistance to change. We argue that out-group dynamics is necessary for polarization, especially in fully connected networks. To illustrate this, we plot the evolution of issue stances by varying $\beta$ relative to $\alpha$ (using values of $\alpha$ estimated for all users in Table 1). The trajectories are shown in Appendix Figs. 7 and 8. When considering only in-group love (homophily), with $\alpha \neq 0$ and $\beta = 0$, we find that liberals and conservatives reach near-consensus or remain weakly polarized (Figs. 7(d) and 8(d)). When group differences disappear, out-group attitude shifts to out-group love ($\beta \to -\alpha$), both groups converge to consensus, becoming pro-masking (or pro-lockdowns) (Figs. 7(a-c) and 8(a-c)). Conversely, as out-group animosity grows ($\beta \to \alpha$), groups diverge along ideological lines. In addition, in-group love is a stabilizing form. In its absence ($\alpha = 0$ and $\beta \neq 0$) both groups converge to $\theta^B(t) = \theta^R(t) \to 0.5$, indicating half the group holds one belief and half the other. Even if both groups started out with a majority of members supporting masking (or lockdowns), without in-group love they evolve into this undecided state. However, if the majority of one group supported masking and only a minority of the other group did ($\theta^B(0) > 0.5$ and $\theta^R(0) < 0.5$), under sufficiently high $\beta$ we would still witness polarization.

## 5.6 State-level Variation of Parameters

We uncover systematic geographic differences in the associations between affective polarization and ideology. Given user's inferred state-level location (determined with Carmen [7]), we separately estimate affective polarization parameters for each state across both issues. While user nodes are restricted to their respective states, their exposures are not limited i.e., we assume that they are exposed to users from across the U.S. We then examine the relationship between a state's conservatism (measured by Republican vote share in the 2020 Federal elections) and three parameters—$\alpha$, $\beta$, and $\delta$—for both issues. Appendix Figure 14(a-c) presents the correlation between these parameters and vote share for masking, and Appendix Figure 14(d-f) for lockdowns. For the masking issue, we find that users in more conservatives states are increasingly less influenced by their in-group and more influenced by their out-group ($\alpha$ falls but $\beta$ rises with Republican vote share), and they become more susceptible to social influence ($\delta$ decreases with Republican vote share). In contrast, for lockdowns, users in more conservative states are more influenced by the in-group ($\alpha$ increases) and less

susceptible to social influence ($\delta$ increases). The issue-specific geographic differences in out-group hate and in-group love emphasize the importance of accounting for both parameters when assessing polarization.

## 6 Conclusion

We present a model of opinion polarization in an emotionally divided society, as well as a framework to estimate in-group love and out-group hate from data. Unlike previous models [2, 3, 6] that relied on homophily, our approach demonstrates that emotional dynamics can polarize even a fully mixed society, in contrast to existing models of polarization based on partisan sorting.

We empirically validate the model using real-world data from discussions of contentious issues during the COVID-19 pandemic, such as masking and lockdowns. After calibrating the model by estimating its three parameters from data, we are able to reproduce the observed levels of polarization and division on each issue. The model captures the sharp ideological divides between liberals and conservatives, and the pronounced gaps among politically active partisans, reflecting real-world trends. Results demonstrate the model's robustness in capturing polarization dynamics and offer a quantitative explanation of how societal divisions can emerge.

Interestingly, in contrast to previous works that demonstrated that out-group hate between political parties in the U.S. has exceeded in-group love [8, 11], our findings indicate that out-group hate is less of a force than in-group love. Had it been weaker still, or group differences did not exist, our model predicts consensus would be achieved on both issues.

***Future Work and Limitations*** Our work opens important avenues for future research. While the proposed model and estimation framework does not assume any specific graph structure, our theoretical analysis and parameter estimation assumed a fully connected graph due to difficulties in collecting social media data. Future studies could relax this assumption and analyze the polarization dynamics and estimate parameters in more structurally rich, sparse networks. It is worth noting that the estimated trajectories were slightly overestimated due to this simplifying assumption. Additionally, robustness of parameter estimation to the choice of the time interval has to be explored.

Our findings emphasize the crucial role of interactions between identity (e.g., ideology) and belief formation. Further research should explore intersectional identities, multi-way interactions, and diverse issue positions within a pluralistic society.We focused on two polarizing issues during the Covid-19 pandemic, which were driven by urgent circumstances. However, traditional wedge issues such as abortion rights, gun control, LGBTQ+ rights, immigration, and racial/social justice have polarized over longer periods. Assessing the applicability of our models and empirical frameworks to these issues could be key to ensuring their generalizability. Despite the validation accuracy of our estimates for issue positions and ideology, they may still be inconsistencies. Relying on large language model for these classifications may introduce biases that could contribute to these inconsistencies.

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

## A Additional Results

Figs. 7 and 8 show the trajectories of issue stances when we vary $\beta$ relative to $\alpha$ while using values of $\alpha$ estimated for all users in Table 1. Fig.14 (a-c) shows the correlation between a state's share of Republican voters in the 2020 Federal elections and the three estimated parameters - $\alpha, \beta, \delta$ for the issue of masking. Fig.14 (d-f) highlights the same for the issue of lockdowns.

## B Additional Details about the Model

### B.1 Formal Definitions of Peer Influence

The formal definitions of the two peer influence measure are given below.

*Definition 1 (Net number of individuals with a stance normalized by degree).* Let,

$$
\begin{aligned}
d_k^{in,0}(v) &= \sum_{(v,u)\in E} \mathbb{1}(R(u) = R(v) \wedge H_k(u) = 0) \\
d_k^{in,1}(v) &= \sum_{(v,u)\in E} \mathbb{1}(R(u) = R(v) \wedge H_k(u) = 1) \\
d_k^{out,0}(v) &= \sum_{(v,u)\in E} \mathbb{1}(R(u) \neq R(v) \wedge H_k(u) = 0) \\
d_k^{out,1}(v) &= \sum_{(v,u)\in E} \mathbb{1}(R(u) \neq R(v) \wedge H_k(u) = 1)
\end{aligned}
\tag{7}
$$

denote the number of in-group and out-group neighbors of $v$ who have stance-0 and stance-1 at time $k$ on graph $G = (V, E)$, and let $d(v) = d_k^{in,0}(v) + d_k^{in,1}(v) + d_k^{out,0}(v) + d_k^{out,1}(v)$ be the degree of node $v$. Then, peer influences are defined as,

$$
\Delta_k^{in,1}(v) = \frac{d_k^{in,1}(v) - d_k^{in,0}(v)}{d(v)}, \qquad \Delta_k^{in,0}(v) = -\Delta_k^{in,1}(v, 1)
$$

$$
\Delta_k^{out,1}(v) = \frac{d_k^{out,1}(v) - d_k^{out,0}(v)}{d(v)}, \quad \Delta_k^{out}(v, 0) = -\Delta_k^{out}(v, 1).
$$

*Definition 2 (Net fraction of neighbors in each group with a stance).* Consider the notation in Eq. 7. The peer influences are defined

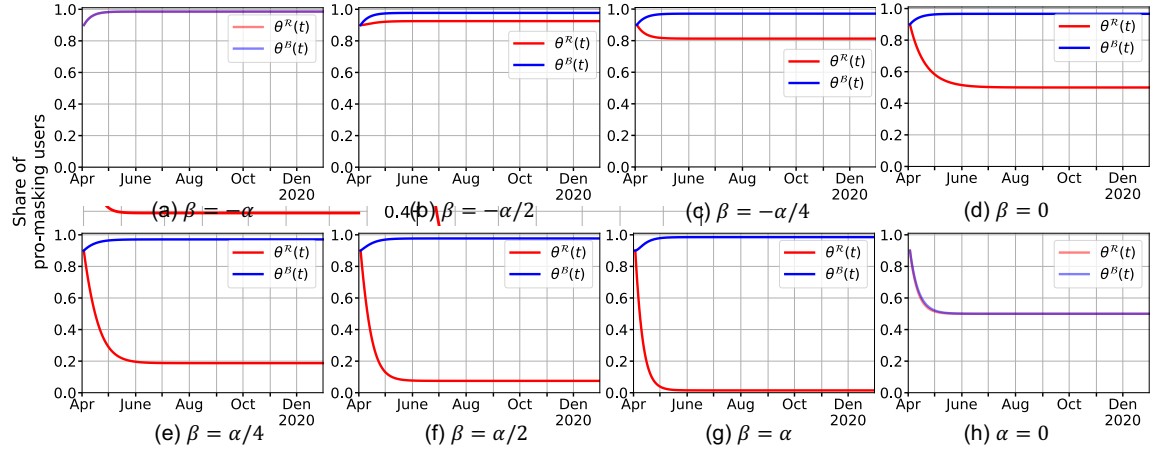

Figure 7: Estimating trajectories of pro-masking users by ideology for varying values of $\beta$ relative to $\alpha$.

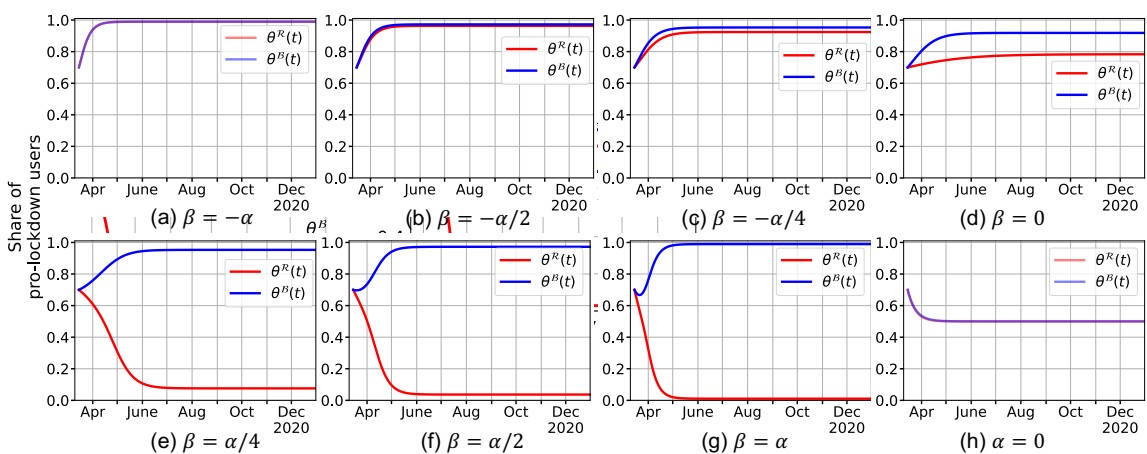

Figure 8: Estimating trajectories of pro-lockdown users by ideology for varying values of $\beta$ relative to $\alpha$.

as,

$$\Delta_k^{in,1}(v) = \frac{d_k^{in,1}(v) - d_k^{in,0}(v)}{d_k^{in,1}(v) + d_k^{in,0}(v)}, \qquad \Delta_k^{in,0}(v) = -\Delta_k^{in,1}(v,1)$$

$$\Delta_k^{out,1}(v) = \frac{d_k^{out,1}(v) - d_k^{out,0}(v)}{d_k^{out,1}(v) + d_k^{out,0}(v)}, \qquad \Delta_k^{out}(v,0) = -\Delta_k^{out}(v,1).$$

Fig. 9 shows how the transition probability $p_{X_k,k}(1|0)$ in Eq. (3) varies as a sigmoid function of the linear combination of the peer influence measures. Consequently, the log-odds (log ratio of switching from 0 to 1 vs. not swicthing) varies linearly.

## B.2 Comparison of Dynamics under Two Influence Measures

Fig. 10 and Fig. 11 show how the dynamics differ under the two influence measures (net number of individuals with a stance normalized by degree and net fraction of neighbors in each group with a stance).

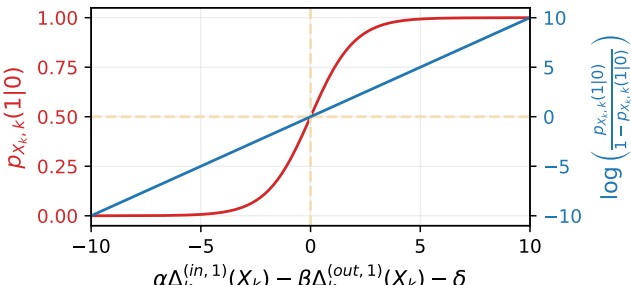

Figure 9: The probability that a random node with stance-0 switches to stance-1 at time $k$ varies as a logistic function (red line) of the $\alpha\Delta_k^{in,1}(X_k) - \beta\Delta_k^{out,1}(X_k) - \delta$ as specified in Eq. 3. Consequently, the log-odds of switching stances vary linearly with $\alpha\Delta_k^{in,1}(X_k) - \beta\Delta_k^{out,1}(X_k) - \delta$ (blue line).

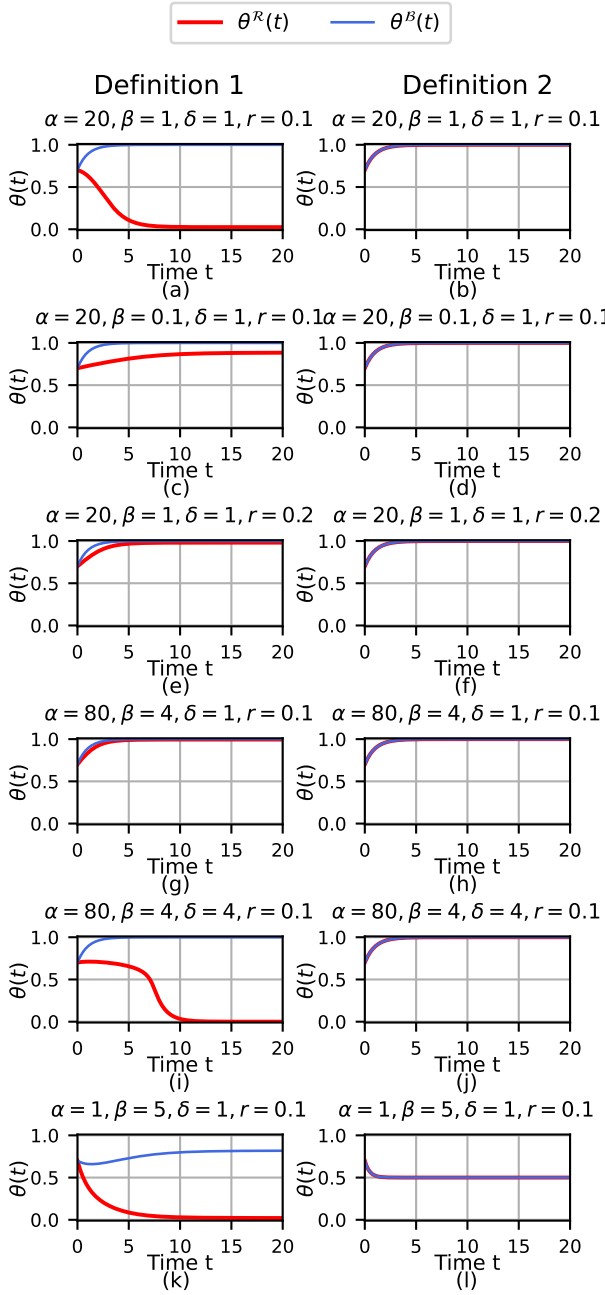

Figure 10: Example trajectories of the $\theta(t) = \left[\theta^{\mathcal{B}}(t), \theta^{\mathcal{R}}(t)\right]$ on a fully connected graph (based on Eq. (4)) under the two definitions (the two columns) of peer influence for various parameter configurations (the four rows). In each case, it is assumed that $\theta^{\mathcal{B}}(0) \neq \theta^{\mathcal{R}}(0)$ i.e., both groups initially have different prevalence of the dynamic attribute.

## B.3 Algorithm for Estimating Parameters

Algorithm 1 outlines the logistic regression approach for estimating the parameters of the proposed model. A dummy variable $J$ is used

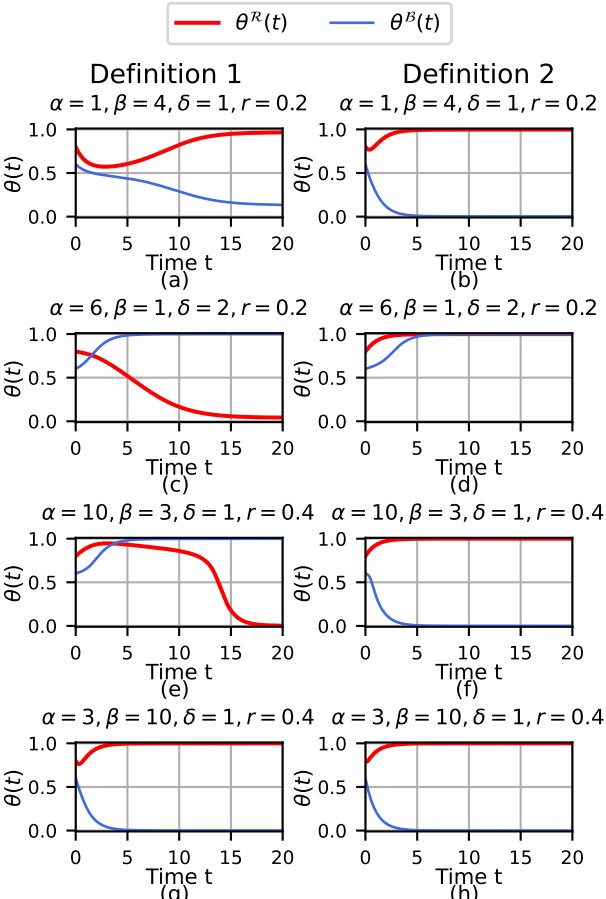

Figure 11: Example trajectories of the $\theta(t) = \left[\theta^{\mathcal{B}}(t), \theta^{\mathcal{R}}(t)\right]$ on a fully connected graph (based on Eq. (4)) under the two definitions (the two columns) of peer influence for various parameter configurations (the four rows). Unlike Fig. 10, the two groups initially have different prevalences of the dynamic attribute.

to denote whether the transition occurred or not, and that dummy variable is treated as the dependent variable.

## C Measuring Exposure Through Tweets

In the main manuscript, we assume that users are exposed to other users, meaning they track how many users hold a particular issue position. Alternatively, one could assume that users track the number of text instances (tweets) associated with a particular group identity and expressing an issue position, rather than the number of users. As an alternative approach to modeling exposures, we modify *Definition 1* as discussed earlier to:

*Alternate Definition. Net number of tweets with a stance normalized by the total number of tweets:* $\Delta_k^{'in,1}(v)$ is defined as the difference between the number of in-group tweets with stance-1 and stance-0, normalized by the number of in-group tweets, and

---

**Algorithm 1:** Estimating $\alpha$, $\beta$, and $\delta$ via Logistic Regression

**Data:** The influence measures $(\Delta_{k_i}^{in,1}(v_i), \Delta_{k_i}^{out,1}(v_i),$
$\Delta_{k_i}^{in,0}(v_i), \Delta_{k_i}^{out,0}(v_i))$ and stances over two adjacent
time instants $(H_{k_i}(v_i), H_{k_i+1}(v_i))$ for nodes
$v_i, i = 1, 2, \ldots, n$

**Result:** Estimates $\hat{\alpha}, \hat{\beta}, \hat{\delta}$

$J \leftarrow [\,] \ X_{in} \leftarrow [\,] \ X_{out} \leftarrow [\,]$

**for** $i = 1, 2, \ldots, n$ **do**

    **if** $H_{k_i+1}(v_i) \neq H_{k_i}(v_i)$ **then**

        $J[i] = 1;$

    **end**

    **else**

        $J[i] = 0;$

    **end**

    **if** $H_{k_i}(v_i) = 0$ **then**

        $X_{in}[i] = \Delta_{k_i}^{in,1}(v_i);$

        $X_{out}[i] = \Delta_{k_i}^{out,1}(v_i);$

    **end**

    **else if** $H_{k_i}(v_i) = 1$ **then**

        $X_{in}[i] = \Delta_{k_i}^{in,0}(v_i);$

        $X_{out}[i] = \Delta_{k_i}^{out,0}(v_i);$

    **end**

**end**

**return** *Logistic regression fit via maximum likelihood:*

$\log\left(\frac{P(J=1)}{1-P(J=1)}\right) = \beta_0 + \beta_1 X_{in} + \beta_2 X_{out}$

Extract parameters: $\hat{\alpha} := \beta_1, \hat{\beta} := -\beta_2, \hat{\delta} := -\beta_0;$

---

**Table 2: Logistic Regression Parameters with exposure as a measure of tweets.**

| Issue | | All users | Pseudo-R2 | Partisans | Pseudo-R2 |
|---|---|---|---|---|---|
| | $\alpha$ | 3.85±0.008 | | 5.27 ± 0.019 | |
| Masking | $\beta$ | 0.08±0.005 | 0.31 | 0.41± 0.005 | 0.38 |
| | $\delta$ | 0.62±0.003 | | 0.28 ± 0.001 | |
| | $\alpha$ | 3.80 ± 0.017 | | 5.03± 0.028 | |
| Lockdowns | $\beta$ | 0.70± 0.014 | 0.18 | 1.22 ± 0.023 | 0.27 |
| | $\delta$ | 0.78 ± 0.004 | | 0.58 ± 0.008 | |

$\Delta_k^{'in,0}(v) = -\Delta_k^{'in,1}(v)$. The quantities for the out-group are defined similarly. For example, let us assume a blue node $v$ is has seen 70 out of 100 blue neighbors' tweets (in-group tweets) to be pro-masks (stance 1) and 7 out of 10 red-neighbors' tweets (out-group tweets) to be pro-masks. Under this first definition of influence, we get $\Delta_k^{'in,1}(v) = (70-30)/110 = 40/110, \Delta_k^{'in,0}(v) = -40/110$ and $\Delta_k^{'out,1}(v) = (7-3)/110 = 4/110, \Delta_k^{'out,0}(v) = -4/110$. We leverage this alternative definition to re-estimate the parameters $\alpha, \beta$ and $\delta$. The results of this estimation are shown in Table 2. Furthermore, the trajectories of pro-masking and pro-lockdown users with liberal and conservative ideologies estimated using these parameters are shown in Figs. 7 and 8. We observe that the estimated trajectories

closely resemble those in Figs. 5 and 6 suggesting that whether we measure exposure in terms of users or tweets, the resulting trajectories are similar.

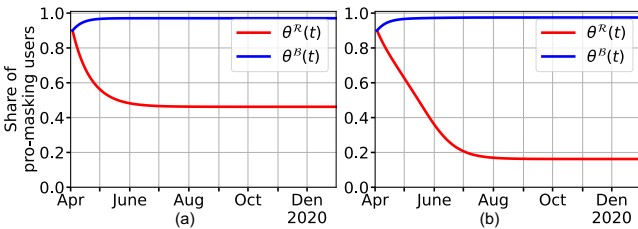

**Figure 12: Plotting trajectories of issue positions on masking for (a) all users and (b) political partisans with exposure measured through tweets.**

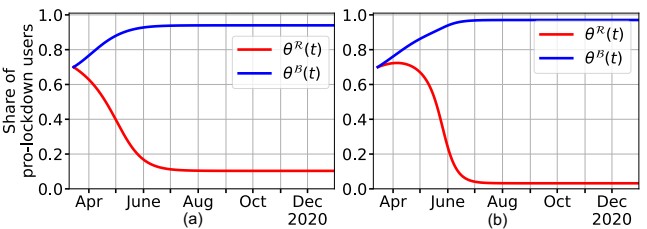

**Figure 13: Plotting trajectories of issue positions on lockdown mandates for (a) all users and (b) political partisans with exposure measured through tweets.**

## D  Extending the Model beyond Two Parties

The model described in Sec. 3 and its analysis for fully connected graphs in Sec. 3.2 can be extended to contexts where there are more than two parties to account for more fine-grained social divisions. For example, one could partition the US population into five non-overlapping groups based on their political ideological leaning: hard-left, left, moderate, right, hard-right. To model such systems, let $A_{ij}$ be the emotion of group $i$ towards group $j$ (where $i, j \in \{1, 2, \ldots, N\}$ and $N$ be the total number of groups). If $A_{ij} > 0$ (resp. $A_{ij} < 0$), then the individuals of group $i$ have a positive (resp. negative) emotion towards group $j$. Then, the transition probabilities of a random node chosen at time $k$, $X_k$, can be expressed as

$$p_{X_k,k}(1|0) = \frac{1}{1 + \exp\left[-\left(\sum_{j=1}^{N} A_{ij}\Delta_k^{j,1}(X_k) - \delta\right)\right]}$$
$$p_{X_k,k}(0|1) = \frac{1}{1 + \exp\left[-\left(\sum_{j=1}^{N} A_{ij}\Delta_k^{j,0}(X_k) - \delta\right)\right]}, \tag{8}$$

where $\Delta_k^{j,1}(X_k), \Delta_k^{j,0}(X_k)$ denote measures that quantify the prevalence of stance-1 and stance-0, respectively, among the neighbors of $X_k$ that belong to group $j$.

Let $\theta^{(i)}(t)$ denote the fraction of individuals in group-$i$ with stance-1, $r_i$ denote the number of group-$i$ individuals as a fraction of the population and $\delta_i$ denote the inertia of group-$i$ individuals.

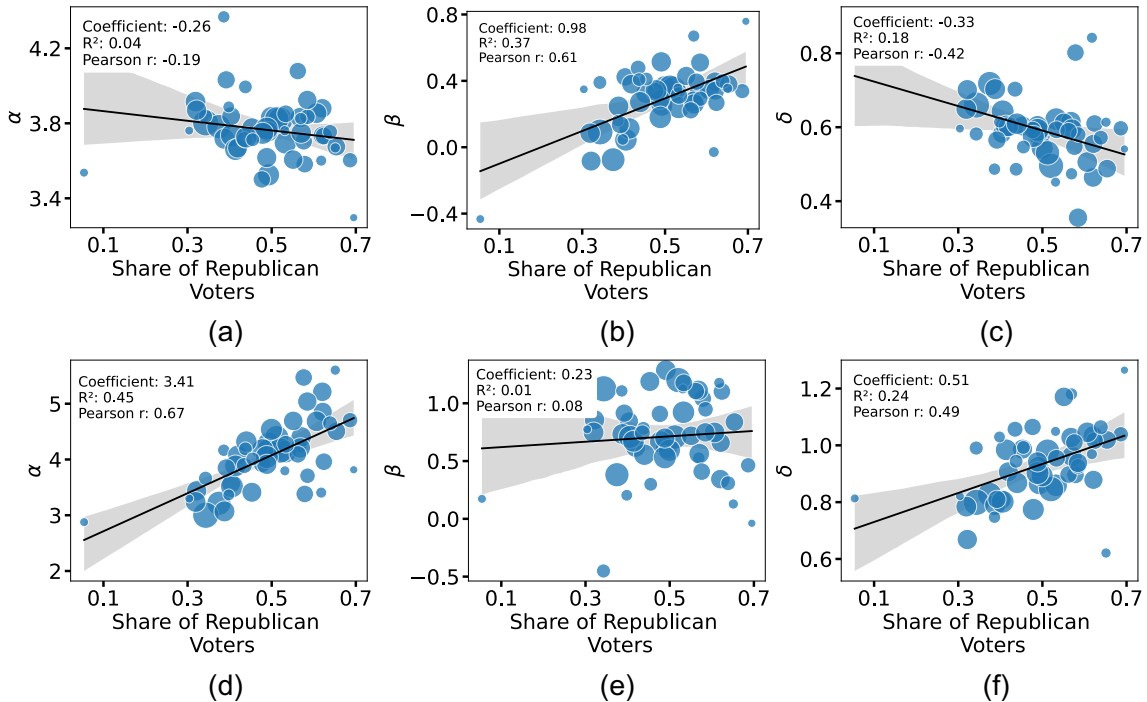

**Figure 14: The model parameters $\alpha, \beta, \delta$ for individuals in each state plotted against the state's 2020 Republican vote share: subplots (a-c) show results for masking and (d-f) for lockdowns.**

Then, the mean-field dynamics of the multi-party system (using the first influence measure) can be expressed as,

$$\frac{d\theta^{(i)}(t)}{dt} = (1-\theta^{(i)}(t)) \cdot p_{01}^{(i)}(t) - \theta^{(i)}(t) \cdot p_{10}^{(i)}(t), \ i = 1, \ldots, N, \quad (9)$$

where,

$$p_{01}^{(i)}(t) = \frac{1}{1 + \exp\left(-\left(\sum_{j=1}^{N} A_{ij} r_j (2\theta^{(j)}(t) - 1) - \delta_i\right)\right)}$$

$$p_{10}^{(i)}(t) = \frac{1}{1 + \exp\left(-\left(\sum_{j=1}^{N} A_{ij} r_j (1 - 2\theta^{(j)}(t)) - \delta_i\right)\right)}. \quad (10)$$

Similar to the two party system, the differential equation Eq. (8) which governs the dynamics in multi-party populations can be numerically estimated. Two example scenarios under the first influence measure (net number of individuals with a stance normalized by degree) are shown in Fig. 15: Fig. 15(a) corresponds to a situation where the two ideologically extreme groups (hard-left and hard-right) are driven more by in-group love and out-group hate compared to the scenario in Fig. 15(b). Additionally, two ideologically extreme groups show some amount of positive emotion towards the moderates in Fig. 15(a) whereas they are indifferent to the moderates in Fig. 15(b). Consequentially, the hard left aligns with the remaining groups in Fig. 15(a) due to the positive emotion towards the largest group, moderates, and the highly negative feeling towards hard-right while the hard-right converges to the opposite stance. Fig. 15(b) illustrates a horseshoe effect [25] where the political extremes end up uniting with each other due to their large animosity towards the more moderate groups.

Thus, the multi-party model can capture a richer array of phenomena compared to the two-party systems even on a fully connected network. As we will subsequently see, such phenomena are observed in stances related to real-world issues, and their parameters can be identified by relying on the proposed model.

## E Additional Details

### E.1 Correlation of Stance and Ideology with Survey Data

Table 3 presents example tweets along with their corresponding stances for each issue. Fig. 16 shows the correlation between the state's 2020 Republican vote share in the Federal elections and the share of state's Twitter users who were identified as conservative by the ideology detection classifier. The Pearson's correlation of $r = 0.86$ with a $R^2$ value of 0.74 indicate that the classifier is reliable. Additionally, Fig. 17 shows the correlation between the share of state's New York Times masking survey [3] respondents who said they'd favor masking sometimes, frequently or always and the share of state's Twitter users who were identified to be pro-masking. The survey was administered from July 1, 2020 to July 14, 2020. We focus on masking tweets from the same period to assess the relationship between the survey responses and Twitter estimates. The Pearson's correlation of $r = 0.63$ with a $R^2$ value of 0.39 validates the stance classifier's performance.

---

[3]https://github.com/nytimes/covid-19-data/tree/master/mask-use

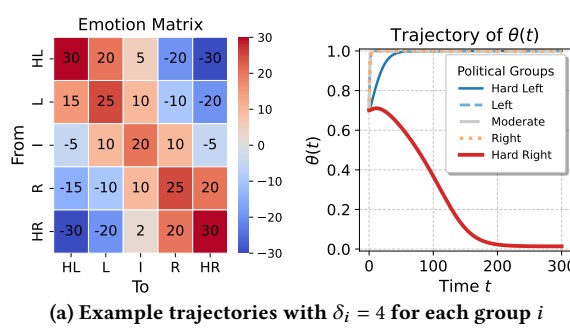

(a) Example trajectories with $\delta_i = 4$ for each group $i$

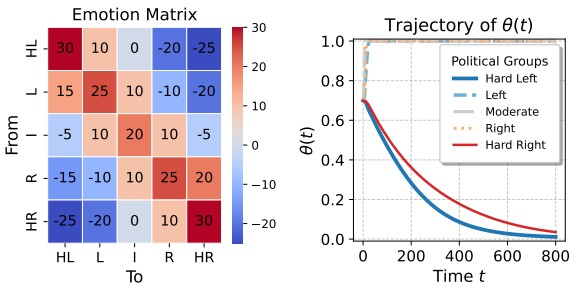

(b) Example trajectories with $\delta_i = 4$ for each group $i$

Figure 15: Comparison of two scenarios (under first influence measure) of an affectively polarized multi-party population with group sizes 5% (HL), 25% (L), 40% (I), 25% (R), 5% (HR). Fig. 15(a) corresponds to a smaller inertia larger in-group love and out-group hate values for the ideologically extreme groups (HL, HR) compared to the scenario in Fig. 15(b).

| Issue | Example Tweet | Stance |
|---|---|---|
| Masking | Wait we're supposed to wear masks? Hell no! I don't wanna smell my own breath all day. what if I had liver and onions for breakfast? | Anti-masking |
| | Masks help stop the spread of coronavirus – the science is simple and I'm one of 100 experts urging governors to require public mask-wearing | Pro-masking |
| | It's hard because I'm not sure whether to trust a rando twitter account versus the cdc director when it comes to the wisdom of masks | Neutral |
| Lockdowns | Every day the left's hypocrisy is on display. Believe all women.... unless they are accusing Democrats. you can't come out of your house. Protesting against the lockdown is wrong and people will die! a protest against racism is ok and it's more important than social distancing. | Anti-lockdowns |
| | Cancel everything. Put us back on lockdown. Cut another check and let try again in a couple months. That's my opinion… | Pro-lockdowns |
| | RT @account: The supreme court just banned covid restrictions on attendance at synagogues and churches. They voted 5-4 | Neutral |

Table 3: Examples of tweets expressing a stance on masking and lockdowns issues.

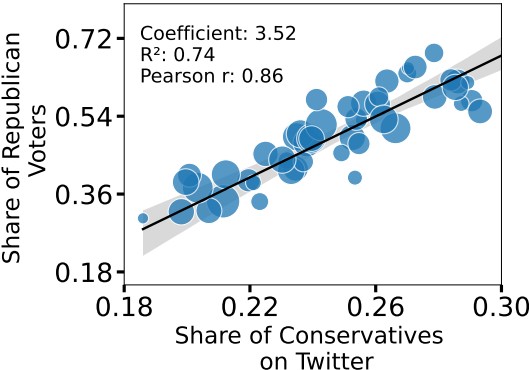

Figure 16: Correlation between state-level Twitter ideology estimates and 2020 Republican vote share in Federal elections.

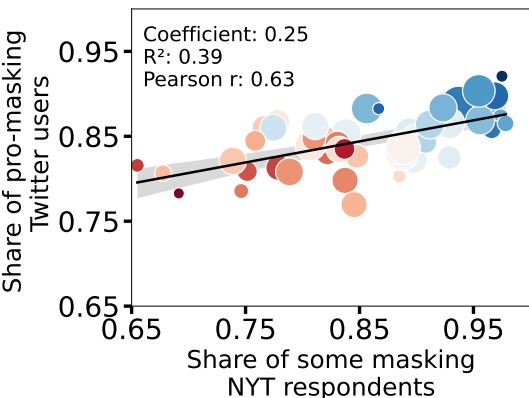

Figure 17: Correlation between state-level New York Times masking survey and Twitter estimates of user stances.

## E.2 Additional Details on Model Calibration

We formalize the estimation of parameters $\alpha, \beta, \delta$, as a supervised learning task. More specifically, we model the shifts in user stances on masking and lockdowns for a user $X$ from $H_t(X)$ to $H_{t+1}(X)$ between two consecutive intervals, $t$ and $t+1$ (where $t = 0,1,2,...$) as a feature of stances of users in the in- and out-group neighborhoods of $X$. We denote the fraction of users who are liberal and pro-masking (resp. lockdowns) and fraction of users who are conservative and pro-masking (resp. lockdowns) at (continuous) time $t$ as $\theta^B(t)$ and $\theta^R(t)$ respectively. Using *Definition 1* (in Sec. 3.1) we quantify for each time period $t$ $\theta^B(t)$ as the number of active liberal users who support the issue at hand, normalized by the total number of *active* users in that period. Similarly, $\theta^R(t)$ represents the number of active conservatives in period $t$ who support the issue, also normalized by the total number of active users in period $t$. Similarly, we define $\theta^{B'}(t)$ as the number of liberals who oppose the issue, and $\theta^{R'}(t)$ as the number of conservative users who oppose the issue, both normalized by the total number of active users in period $t$. We can then model the transition probabilities as:

$$p_\theta^B(1|0) = logit^{-1}\left(\alpha\left(\theta^B(t) - \theta^{B'}(t)\right) - \beta\left(\theta^R(t) - \theta^{R'}(t)\right) - \delta\right),$$

$$p_\theta^B(0|1) = logit^{-1}\left(\alpha\left(\theta^{B'}(t) - \theta^B(t)\right) - \beta\left(\theta^{R'}(t) - \theta^R(t)\right) - \delta\right),$$

$$p_\theta^R(1|0) = logit^{-1}\left(\alpha\left(\theta^R(t) - \theta^{R'}(t)\right) - \beta\left(\theta^B(t) - \theta^{B'}(t)\right) - \delta\right),$$

$$p_\theta^R(0|1) = logit^{-1}\left(\alpha\left(\theta^{R'}(t) - \theta^R(t)\right) - \beta\left(\theta^{B'}(t) - \theta^B(t)\right) - \delta\right).$$

We represent $p_\theta^B(1|0), p_\theta^B(0|1), p_\theta^R(1|0), p_\theta^R(0|1)$ using a dummy variable $J$, that indicates transition. We have $J = 1$ when we have a transition from anti to pro-masking (resp. lockdowns) regardless of the group identity ($p_\theta^B(1|0), p_\theta^R(1|0)$). Conversely, we have $J = 0$ when we have a transition from pro to anti-masking (resp. lockdowns) regardless of the group ($p_\theta^B(0|1), p_\theta^R(0|1)$).

## E.3 Identifying Geolocation

Location information for tweets is available for a subset of tweets through the coordinates and place fields within the tweet object. However, this data is present in less than 5% of the tweets in our dataset. To address this, we used Carmen [7] [4], a geo-location inference tool for Twitter data, to assign tweets to U.S. locations. Carmen utilizes information present in the user's bio, in addition to the place and coordinates fields of the tweet object, to infer location. The location object provided by Carmen includes details like the country, state, and county of the tweet. A manual review confirmed the method's effectiveness in identifying a user's home state.

---

[4]https://github.com/mdredze/carmen-python

