# OpenReview forum: "In-Group Love, Out-Group Hate: A Framework to Measure Affective Polarization via Contentious Online Discussions"
_ACM.org/TheWebConf/2025/Conference — WWW 2025 Oral_

### Official Review · Reviewer_zecR · 2024-11-15

**Novelty:** 5
**Technical Quality:** 6

**Review:**

Thank you for the opportunity to review this manuscript. The work presents several interesting findings, though there are some areas that could be strengthened.

1) The clear presentation of methods and theoretical framework in the early sections provides a smooth foundation for readers to follow the work.

2) The figures are well-designed and effectively captioned, making complex data easily interpretable.

3) The authors successfully integrate the concept of out-group animosity into the “more established” framework of in-group favoritism, offering a more comprehensive analytical approach.

4) The literature review is brief yet comprehensive. The authors provide an honest and thoughtful discussion of their methodology's limitations. However, I find that the authors could have covered more existing structural based methods that have been specifically designed for affective polarization, such as https://www.nature.com/articles/s42005-023-01467-8.

5) Figure 14 in the Appendix effectively bridges theoretical modeling with practical applications, providing a compelling visualization of how model parameters relate to intepretable real-world contexts.

6) The authors successfully employ and validate multiple machine learning models for various detection tasks, demonstrating the robustness of their approach.

7) While the work builds upon established influence dynamics models, it makes a valuable contribution by expanding the model's capability to capture a broader range of scenarios in polarized enviroments.

8) While the authors acknowledge the limitation explicitly, expanding the methodology to incorporate graph structure could provide additional insights into interaction patterns.

9) The analysis could benefit from a comparative study using non-contentious online discussions as “a control group”, as done here https://dl.acm.org/doi/abs/10.1145/3140565. Such a benchmark would help validate that affective factors, rather than general discussion dynamics, are indeed driving the observed interaction patterns.

10) Authors could have compared more extensively their method to ones that measure polarization based on homophily (such as https://dl.acm.org/doi/10.1145/3512962), although some discussion was included. I think there would have been great opportunity to discuss deeper the interplay between homophily and out-group animosity, as separating these can be quite hard from network data.

**Questions:**

I have two crucial questions on the overall contribution, and then some smaller “inquiries”.

1) Why are you talking about in-group love and out-group hate instead of in-group favoritsm and out-group animosity, which are more prevalent in the literature? You actually use these terms as well in the paper towards the end of paper (e.g. line 786). Love is a strong emotion and I am not sure whether the love is really the underlying factor driving in-group favoritsm.

2) While you derived intuitive interpretations for the alpha, beta and gamma values, how should these really be interpreted? We saw that distinct values with equal ratios can lead to very different scenarios. I understand the flexibility and “creativity” of the model comes with costs, but given these “non-linear” behavior of these parameters in the context of polarization, how can we rank distinct systems? Should we use the final “partisan gaps” or can these inferred parameters be used somehow to distill a meaningful score?

LINE 82: However, these methods do not generalize well due to challenges of collecting network data, and they fail to explain divergence of partisan opinions in fully connected networks.

What challenged are you exactly referring to here? And why they fail?

LINE 122: By rapidly identifying emotionally polarized issues on social media through empirical estimation of in-group love and out-group hate, we can create more effective strategies that foster more constructive dialogue.

I find this a slightly lazy way to express the motivation of studying affective polarization. How identifying emotionally polarized issues online rapidly exactly help us create strategies to foster constructive dialogue? We know that media polarizes and that social media platforms do not necessarily have the incentives to reduce online polarization.

FIGURE 1: Why in majority of the plots the size difference between minority and majority groups are so large (r being mostly 0.1-0.2)? Do trends look different with almost equal sized chambers?

LINE 595: More specifically, we use tweets as input to the model with the prompt - What is the stance expressed towards masking (resp. lockdowns mandates) in the following tweet?

How do you ensure the consistency of the stochastic LLM outputs? How do you handle differently formatted outputs or do you apply some structural outputs technique to get the annotations?

LINE 605: Masking stances are also highly correlated at state level with off-line masking survey conducted by New York Times (Pearson 𝑟 = 0.63, see Appendix E.1).

Correlated indeed, but highly correlated?

I enjoyed reading the paper - I hope you find my comments constructive!

**Reviewer Confidence:**

4: The reviewer is certain that the evaluation is correct and very familiar with the relevant literature

**Scope:**

4: The work is relevant to the Web and to the track, and is of broad interest to the community

---

### Official Review · Reviewer_Sh3Q · 2024-11-26

**Novelty:** 3
**Technical Quality:** 4

**Review:**

This paper applied the concept of in-group favouritism and out-group animosity from [1] to analyse polarization in online discussion regarding COVID-19. The paper introduced a logistic regression using the proportion of stances within in-group and out-group neighbours as variables; the weight $\alpha$ and $\beta$ indicate how the person's future stance is affected by their neighbours. The findings show that in-group love has a higher effect than out-group hate. The paper is easy to follow and has a clear motivation. However, the method has limited technical contribution towards the area; the logistic regression consists of two variables that have oversimplified the issue to some extent, and the utilisation of spatial-temporal information from the social network is quite limited.

[1] Nettasinghe, B., Percus, A. G., & Lerman, K. (2024). Dynamics of Affective Polarization: From Consensus to Partisan Divides. arXiv preprint arXiv:2403.16940.

**Questions:**

- How do the results and findings compare with the methods in related work from line 159?
- How does this method compare with graph neural network-based methods on this dataset?

**Reviewer Confidence:**

3: The reviewer is confident but not certain that the evaluation is correct

**Scope:**

4: The work is relevant to the Web and to the track, and is of broad interest to the community

---

### Official Review · Reviewer_UBiz · 2024-11-27

**Novelty:** 3
**Technical Quality:** 3

**Review:**

This manuscript develops an opinion formation model on networks where the opinions of nodes change based on the opinions of their network neighborhood. The model includes explicit group memberships that can bias how much the neighbors affect the opinions. The authors then derive rate equations for fully connected networks with large population sizes. The authors then fit their analytical solution to data on masking and lockdowns for liberal and conservative Twitter users. The authors observe qualitatively somewhat similar time series curves for the models and the data.

Strengths:
- Nice data set with large-scale data on a recent interesting case
- Authors are able to infer the opinions and political leaning of users

Weaknesses:
- There is a major gap between what authors claim their measures do and what they actually measure. The authors say that they measure affective polarization (in-group love and out-group hate), but they model influence within and across groups. These are clearly not the same thing, nor it is justified in any way in the manuscript. I can be influenced by someone without any emotional link or vice versa. There can of course be some correlation between the two. I don’t see how the model can distinguish affective polarization from different types of polarizations as the authors imply.
- The authors don’t engage with the literature on measuring polarization (even the with literature of measuring it in social media)
- The authors completely ignore the literature on opinion formation models, which their model is an example of. There are even works that model network-based polarization in Twitter, such as https://link.springer.com/article/10.1007/s10618-017-0527-9
- The authors use a fully mixed network model, which is essentially the type of model that was considered in the past without networks. Since then, there is a literature that shows how the network structure can me important for this type of dynamics. Especially since there are two groups in the system the underlying network probably reflects that and can have a large impact. On the other hand, the bias terms could  be thought to take the role of the network with groups but in this case the model could become similar to an opinion formation model with network groups(?)
- I’m not convinced of the validation of the model. Authors say they capture the polarization dynamics accurately, but looking at the results it seems clear that they are not accurate but perhaps qualitatively similar. There is nothing wrong with this result, but it should not be sold as something it is not.

**Questions:**

See review.

**Reviewer Confidence:**

3: The reviewer is confident but not certain that the evaluation is correct

**Scope:**

4: The work is relevant to the Web and to the track, and is of broad interest to the community

---

### Official Review · Reviewer_61JV · 2024-12-02

**Novelty:** 3
**Technical Quality:** 4

**Review:**

The paper proposed two models to measure affective polarization on social media: an intuitive discrete choice model that indicates the user’s ideology tendency and a statistical estimation model to estimate in-group love and out-group hate. The paper also uses pandemic-related data as an example for empirical evaluations of the models.

The paper is clearly structured and the result presentation is comprehensive. However, I found several weaknesses in the motivation and evaluation which I believe could improve the contribution of the paper if addressed by the authors:
1. I found the motivation not well-established, firstly, the authors argued the existing method relies largely on survey-based methods. However, I noticed that there is some existing work about using quantitative methods to measure affective polarization. The novelty of the paper could be significantly improved if the authors draw a comparison between the proposed method and existing quantitative measure methods, not just the survey-based method. The following literature is not comprehensive but can be considered:
a)Martínez-España, Raquel, et al. "Methodology for measuring individual affective polarization using sentiment analysis in social networks." IEEE Access (2024).
b)Feldman, Dan, et al. "Affective polarization in social networks." arXiv preprint arXiv:2310.18553 (2023).
c)Borrelli, Dario, et al. "A quantitative and content-based approach for evaluating the impact of counter narratives on affective polarization in online discussions." IEEE Transactions on Computational Social Systems 9.3 (2021): 914-925.
2. In section 2, the authors argued the proposed method is objective, scalable, interpretable, which distinguishes the work from existing ones. However, I found there is lack of elaboration on these features in comparison with existing methods
3. The paper could benefit from a deeper discussion about how in-group love and out-group hate can be used in building a less toxic community on social media without hurting the freedom of speech. I suggest the authors expand this topic with the synthesis of the paper “What Do We Measure When We Measure Affective Polarization?” by Druckman et al.

**Questions:**

I found the section 5.3 confusing. The section is about validating the proposed method in the pandemic-related data. The classification includes two phases: measuring the interaction with political elites and supervised fine-tuning of the LLaMA 3.1-8B Instruct model. However, the authors mentioned that in the dataset, most of the users did not interact with political elites. If I understand correctly, the author used the results of “the follower network-based approach” as the “ground truth” and calculated the Pearson score and f1 score. Can the authors further justify the methodology and include the performance of the follower network-based approach to make the results more convincing?

**Reviewer Confidence:**

3: The reviewer is confident but not certain that the evaluation is correct

**Scope:**

4: The work is relevant to the Web and to the track, and is of broad interest to the community

---

### Official Review · Reviewer_996o · 2024-12-03

**Novelty:** 4
**Technical Quality:** 5

**Review:**

## Strengths
- This is an important substantive topic.
- The predictive power of the model appears to be strong.
- Intuitive generalization to beyond two groups with potentially very interesting results. On this note, I think it’s unwise to relegate this in the appendix, as it is an important contribution that should be expanded on and somehow engaged with more, i.e., beyond Fig 15, with both empirical and simulated results. This concerns a bit how the authors intend to situate this manuscript. (More on this below.)
- Writing is generally clear and reasonably easy to follow despite extensive use of notations.

## Areas of improvement
- It is not immediately clear how the authors want to situate the contribution of this manuscript. The introduction focuses heavily on the potential methodological contribution (without talking as much about the research gaps in the polarization literature), but the emphasis of the results are on the substantive findings and how it relates to existing substantive work on the role of ingroup love and outgroup hate in polarization research (and do not engage as much with the methodological contributions). Additionally, there are a lot of problems or gaps raised about existing studies or the literature, but the manuscript does not actually address these gaps (e.g., emotional dynamics, real time measurement of affective polarization).
- Related to the above point, while the authors try to situate their results in the polarization literature, it does not actually appropriately engage with the literature. For example, the authors argue that their study novelly shows the relative weakness of outgroup hate in determining behavior, which supposedly differs from existing work. But the authors do not explicitly point to studies that actually present outgroup hate as the mechanisms driving policy behavior (while there are many studies on elite-led polarization, which arguably falls more in line with the ingroup love mechanism).
- There are many claims that come out of nowhere, and are not elaborated on, or are not backed up. One example: “It is objective, scalable, interpretable, and compatible with modern tools like LLMs” (lns 203–204). Another example: “existing models of opinion change fail to account for emotional dynamics nor offer methods to quantify affective polarization robustly and in real-time” (abstract).
- Single case (or two very similar cases under the same context, if we think of masking and lockdowns as two cases), which is in many ways a weird situation. It potentially has benefits for what the authors want to do, but this is not clearly explicated. But still, this being a single case makes it difficult for the authors to more strongly demonstrate the empirical validity of their model (e.g., different cases that exhibit very different model parameters), which appears to be high.
- It’s not clear what this has to do with the web other than the data analyzed. In fact, couldn’t this be done with temporal survey data which lets the author estimate population proportions over time? (This is a strength, in general, just not necessarily for the Web Conf).
- The discussion of the discrete choice model dynamics with the simulated data (Figs 1 and 2) is interesting, but it is not sure what the takeaway is. Perhaps this can be done more systematically.

**Questions:**

- Lack of elaboration on how the current paper improves [27]. [27], in fact comes up in this paper more than once, so it is important to know exactly what the improvements are so we have a sense of whether the current manuscript contains enough original contribution.
- More meaningfully discuss the relationship between this study and existing works on polarization beyond mentioning what others have done and what this study does. What do the findings in this paper contribute to the literature? For example, how does it address specific existing studies? (I recognize there is some of this in the manuscript, but I do not think it is sufficient.)

**Reviewer Confidence:**

3: The reviewer is confident but not certain that the evaluation is correct

**Scope:**

2: The connection to the Web is incidental, e.g., use of Web data or API